# Transcriptome Profiles Reveal the Promoting Effects of Exogenous Melatonin on Fruit Softening of Chinese Plum

**DOI:** 10.3390/ijms241713495

**Published:** 2023-08-30

**Authors:** Zhiyu Li, Lu Zhang, Yaxin Xu, Xuemei Zhang, Yanzhou Zhu, Jin Wang, Hui Xia, Dong Liang, Xiulan Lv, Lijin Lin

**Affiliations:** College of Horticulture, Sichuan Agricultural University, Chengdu 611130, China; 2021205022@stu.sicau.edu.cn (Z.L.); zl@stu.sicau.edu.cn (L.Z.); 2021305069@stu.sicau.edu.cn (Y.X.); zhangxuemei030416@163.com (X.Z.); 15182716117@163.com (Y.Z.); 14224@sicau.edu.cn (J.W.); xiahui@sicau.edu.cn (H.X.); liangeast@sicau.edu.cn (D.L.); 10998@sicau.edu.cn (X.L.)

**Keywords:** melatonin, softening, cell wall metabolism, plum fruit, transcriptome analysis

## Abstract

In this study, we investigated the effect of exogenous melatonin (MT) on cell wall metabolism leading to Chinese plum (*Prunus salicina* Lindl.) fruit softening. Exogenous MT treatment increased the endogenous MT content in plum fruits before fruit ripening. However, in mature plum fruits, exogenous MT treatment decreased the fruit hardness, pulp hardness, fruit elasticity, contents of ion-bound pectin, covalently-bound pectin, hemicellulose, and cellulose, and activities of xyloglucan endotransglycosylase/hydrolase and endo-β-1,4-glucanase, and increased the water-soluble pectin content, and activities of pectin methyl esterase, pectin lyase, polygalacturonase, β-galactopyranosidase, and α-L-arabinofuranosidase. Transcriptome analysis revealed that the differentially expressed genes (DEGs) associated with cell wall metabolism in the exogenous MT-treated plum fruits were mainly enriched in the pentose and glucuronate interconversions, phenylpropanoid biosynthesis, cyanoamino acid metabolism, and galactose metabolism pathways. Analysis of these DEGs revealed that exogenous MT treatment affected the expression of genes regulating the cell wall metabolism. Overall, exogenous MT treatment promotes the fruit softening of Chinese plum.

## 1. Introduction

The change in cell wall metabolism is an important indicator of fruit ripening, which plays a critical role in fruit softening [1]. The cell wall is a complex, multilayered dynamic network structure composed of pectin, cellulose, hemicellulose, mineral elements, and a few structural proteins [2]. At the early stage of fruit development, the cell wall is thick but gradually disintegrates, leading to cell expansion as the fruit develops. Subsequently, the cell wall structure is significantly disrupted, the hemicellulose depolymerizes, and intercellular adhesion is significantly reduced at the late stage of fruit ripening [3].

Melatonin (MT) is a tryptophan-indole derivative widely found in plants and animals. It was first extracted from the pineal gland of cattle in 1958, hence the name pinealectin [4]. In vertebrates, MT is synthesized in the pineal gland in four consecutive enzymatic reactions with tryptophan as the precursor [5]. In the human body, MT plays a key role in regulating the biological clock and the sleep–wake cycle, and also has anti-inflammatory, antioxidant, immunomodulatory, analgesic, and anxiety-reducing properties [6]. In plants, there are multiple pathways for MT synthesis; thus, MT synthesis in plants is more complex [7]. Several recent studies have pointed out that MT is important in regulating seed germination, morphogenesis, material metabolism, and resistance to external stresses [8]. In addition, MT influences plant maturation and cell wall metabolism [9].

In fruits, high pectin methyl esterase (PME) and low polygalacturonase (PG) activities significantly increase the accumulation of demethylated pectin, resulting in fruit pulverization [10]. However, treatment with 100 μM of MT maintains a balance between *PpPME* and *PpPG* transcripts, up-regulates some arabinogalactan-like protein family genes, and alleviates cell wall modification by positively regulating the activity of cell wall modifying enzymes and protein synthesis, which alleviates fruit pulverization [11]. In pear fruit, treatment with 100 μM of MT affected the expression of genes related to cell wall degradation, though the effects varied depending on the pear variety [12]. However, there are no systematic studies on the influence of MT on fruit cell wall metabolism.

Ethylene plays an important role in fruit ripening. However, Hu et al. (2017) [13] pointed out that the changes in MT content in bananas are more consistent with the trend of ripening changes than ethylene. Notably, treatment with 50 μM of MT reduces ethylene production by regulating the expression of *MaACO1* and *MaACS1*, which delays the post-harvest ripening of fruits. More importantly, the exogenous MT (100 μM) treatment affects the endogenous hormone level in sweet cherries. Interestingly, the MT trend is consistent with that of jasmonic acid and salicylic acid in plants. At the same time, MT inhibits sweet cherry fruit ripening by crosstalk with cytokinins [14]. In strawberry fruits, MT (1000 μM) increases H_2_O_2_ accumulation and up-regulates *TDC*, *SNAT*, *T1000H*, *ASMT*, and abscisic acid content to promote fruit ripening [15]. Similarly, MT plays an important role during fruit ripening in tomatoes and apples [16,17].

Chinese plum (*Prunus salicina* Lindl.) is a common temperate fruit with a high economic value and is widely loved by consumers, but the content of lignin in the fruit affects its commercial value [18,19]. MT biosynthesis has the same regulatory factors as lignin biosynthesis, and exogenous MT treatment influences plant lignin synthesis [20]. Although previous research has revealed that MT affects the ripening of horticultural crops [15,16,17], there is little information on the influence of MT on fruit cell wall metabolism during fruit ripening. Therefore, this study investigated the effects of exogenous MT application on fruit cell wall metabolism during Chinese plum fruit ripening and used the transcriptome profiles to explore the molecular mechanisms regulating the plum fruit cell wall metabolism following MT treatment.

## 2. Results

### 2.1. Texture Analysis of Plum Fruits

There were no significant changes in plum fruit hardness at 7 and 14 days following exogenous MT treatment compared to the control (Figure 1A). However, the exogenous MT treatment significantly increased the fruit hardness at 21 days and decreased it at 28 days compared to the control. At the same time, the pulp hardness showed a gradual decrease from fruit development to maturity. Notably, the changes in pulp hardness after the exogenous MT treatment was consistent with the changes in fruit hardness (Figure 1B), with significant changes at 21 and 28 days only. The exogenous MT treatment increased the pulp hardness by 7.15% compared to the control at 21 days and decreased it by 14.43% at 28 days. In contrast, the exogenous MT treatment had no significant changes in fruit elasticity at 7, 14, and 21 days, which decreased by 5.68% compared to the control at 28 days (Figure 1C). Therefore, the exogenous MT treatment decreased the hardness and elasticity of plum fruits only at maturity.

### 2.2. Endogenous MT Content in Plum Fruits

The endogenous MT content in plum fruits showed an increasing trend following the exogenous MT treatment (Figure 1D). Specifically, the exogenous MT treatment significantly increased the endogenous MT content by 4.17, 5.26, and 23.61% at 7, 14, and 21 days, respectively, compared to the control. However, there was no significant difference in the endogenous MT content between exogenous MT treatment and control plum fruits at 28 days.

### 2.3. Cell Wall Substance Content in Plum Fruits

The water-soluble pectin in plum fruits decreased during plum fruit development (Figure 2A). Exogenous MT treatment only increased the water-soluble pectin content at 21 and 28 days by 10.86 and 12.27%, respectively, compared to the control. However, there were no significant changes in the water-soluble pectin content at 7 and 14 days between the exogenous MT treatment and the control.

The ion-bound pectin content was lower in the exogenous MT-treated plum fruits than in the control at 28 days (Figure 2B). Similarly, the covalently-bound pectin content was lower in the exogenous MT-treated plum fruits than in the control at 21 and 28 days (Figure 2C). However, the changes in the ion-bound and covalently-bound pectin treated with exogenous MT were insignificant compared to the control. Exogenous MT treatment also decreased the hemicellulose content at 7 and 28 days but had no significant changes at 14 and 21 days compared to the control (Figure 2D). On the contrary, the exogenous MT treatment increased the cellulose content by 6.82 and 9.88% at 7 and 14 days, respectively, and decreased it by 7.31% at 28 days (Figure 2E) compared to the control. Interestingly, the exogenous MT treatment only decreased the lignin content at 7 days, with no significant effects on 14, 21, and 28 days (Figure 2F). Therefore, exogenous MT treatment promoted cell wall degradation in plum fruits at maturity.

### 2.4. Cell Wall Metabolism-Related Enzyme Activity in Plum Fruits

Exogenous MT treatment had no significant effect on the PME activity in plum fruits in the early stage of plum fruit development (Figure 3A). However, at 28 days, the exogenous MT treatment increased the PME activity by 70.76% compared to the control. For the PL activity, exogenous MT treatment significantly decreased its activity at 14 days but increased it at 28 days (Figure 3B). On the contrary, there were no significant differences in the PL activity between the exogenous MT-treated and control plum fruits at 7 and 21 days. The PG and β-GAL activities in exogenous MT-treated plum fruits were higher than the control at 7, 14, 21, and 28 days (Figure 3C,D). Exogenous MT treatment also increased the XTH activity at 7 and 14 days but decreased it at 21 and 28 days (Figure 3E). The α-AF activity in the exogenous MT-treated plum fruits was lower than the control at 7, 14, and 21 days and higher than the control at 28 days (Figure 3F). The EG activity in the exogenous MT treatment fruits was lower than the control at 7 and 28 days, with no significant differences with the control at 14 and 21 days (Figure 3G).

### 2.5. The DEGs Following Exogenous MT Treatment

To further investigate the effect of exogenous MT on the cell wall metabolism during ripening of plum fruits, transcriptome sequencing analysis was used to analyze the samples of plum fruits at 21 and 28 days after treatment. A total of 12 samples (c represents 21 days, and d represents 28 days) were analyzed on the Illumina platform. The sequencing results revealed that the GC content was greater than 45%, and the percentage of Q30 was greater than 94% (Table 1) implying a high sequencing and library construction sample quality. The comparison efficiency of the sample reads with the reference genome ranged between 93.29 and 94.61% (Table 2). The Pearson correlation coefficient further revealed a high correlation between all samples (Figure 4). In addition, 450 DEGs were detected at 21 days, of which 157 were down-regulated, and 302 were up-regulated (Figure 5A). At the same time, 380 DEGs were detected at 28 days, of which 297 were down-regulated, and 83 were up-regulated. Among them, 45 DEGs were detected at both 21 and 28 days (Figure 5B).

### 2.6. Functional Classification of DEGs

The GO enrichment analyses revealed that 350 DEGs were enriched at 21 days and 306 DEGs at 28 days (Table 3). The DEGs at 21 days were enriched in the biological processes, including photosynthesis, light harvesting in photosystem I, protein-chromophore linkage, xyloglucan metabolic process, xyloglucan metabolic process, chromophore linkage, xyloglucan metabolic process, plant-type cell wall organization, cell wall organization, and cell wall biogenesis (Figure 6A). At 28 days, the DEGs were enriched in photosynthesis, suberin biosynthesis, and lipid metabolic processes (Figure 6D). In the cellular components, the DEGs were enriched in the photosystem I, photosystem II, plastoglobule, and cell wall at 21 days (Figure 6B) and photosystem I reaction center, integral membrane components, and cell wall at 28 days (Figure 6E). In the molecular functions, the DEGs were enriched in chlorophyll-binding, xyloglucan: xyloglucosyl transferase activity, and cellulose synthase (UDP-forming) activity at 21 days (Figure 6C), and the transferase activity, transferring acyl groups other than amino-acyl groups, pectinesterase inhibitor activity, among others, at 28 days (Figure 6F).

Subsequently, KEGG pathway enrichment analysis detected 302 and 260 DEGs enriched at 21 and 28 days, respectively (Table 3). At 21 days, the DEGs were mainly enriched in the photosynthesis-antenna proteins, plant–pathogen interaction, and glycerophospholipid metabolism pathways (Figure 7A). At 28 days, the DEGs were mainly enriched in photosynthesis and plant–pathogen interactions (Figure 7B).

### 2.7. Analysis of the Cell Wall Metabolism-Related DEGs

DEGs analysis identified 11 genes related to cell wall metabolism in the exogenous MT-treated group at 21 days. These cell wall metabolism-related DEGs were mainly enriched in the pentose and glucuronate interconversions and phenylpropanoid biosynthesis pathways (Figure 8). PG, galacturan 1,4-alpha-galacturonidase, and UDP-glucose 6-dehydrogenase activities were up-regulated in the pentose and glucuronate interconversions pathway. At the same time, the PME activity inhibitor and PL were up and down-regulated (Figure 8A). Additionally, the expression of cinnamoyl-CoA reductase was down-regulated in the pentose and glucuronate interconversions pathway, while peroxidase was up and down-regulated (Figure 8B).

At 28 days, 16 genes related to cell wall metabolism in the exogenous MT-treated plum fruits were mainly enriched in the pentose and glucuronate interconversions, phenylpropanoid biosynthesis, cyanoamino acid metabolism, and galactose metabolism pathways. Pectinesterase and pectate lyase were down-regulated in the pentose and glucuronate interconversions pathway, and galacturan 1,4-alpha-galacturonidase was up-regulated (Figure 8C). In the phenylpropanoid biosynthesis pathway, beta-glucosidase was up-regulated, cinnamoyl-CoA reductase was down-regulated, and shikimate O-hydroxycinnamoyltransferase was both up-regulated and down-regulated (Figure 8D). Beta-glucosidase was up-regulated in the cyanoamino acid metabolism pathway (Figure 8E). In the galactose metabolism pathway, UDP-sugar pyrophosphorylase and beta-galactosidase were down-regulated (Figure 8F).

### 2.8. qRT-PCR Analysis of the DEGs

The expression patterns of the six and nine DEGs at 21 and 28 days, respectively, screened by qRT-PCR analysis, were consistent with the RNA-seq results (Figure 9). Notably, *PPE8B*, *XTH23*, *At4g24780*, and *GSVIVT0002692001* were up-regulated, and *RCA* and *PME* were down-regulated at 21 days post-MT treatment. At 28 days, *PGIP*, *Xyl2*, *BGLU41*, and *BGLU11* were up-regulated, and *PPE8B*, *RCA*, *BGAL8*, *XTH23*, and *PECS-1.1* were down-regulated in the MT-treated group.

## 3. Discussion

In eggplants, exogenous MT treatment up-regulates the MT synthesis-related genes and the endogenous MT content [21]. In apples, exogenous MT application also increases the endogenous MT content [22]. Increasing endogenous MT content in tomatoes promotes fruit ripening and softening [23]. In various fruits, the exogenous MT treatment can delay or promote fruit ripening by mediating ethylene biosynthesis and signal transduction during fruit ripening [24]. In this study, exogenous MT treatment increased the endogenous MT content in plum fruits 7, 14, and 21 days post-treatment. This implies that the exogenous MT treatment promoted the accumulation of endogenous MT in plum fruits during fruit development, consistent with the results from previous studies [21,22]. However, the exogenous MT treatment had no significant effect on the endogenous MT content in plum fruits at 28 days (maturity period), implying that the endogenous MT content increased during fruit development and decreased during fruit ripening [24].

The changes in the cell wall components are directly related to the changes in the fruit texture [25]. Pectin is one of the important components of the cell wall. Generally, the cell wall hardness is closely related to the pectin structure and chemical modifications [26]. Fruit softening is accompanied by the degradation of the cell wall material, reduction of pectin methyl esterification and long-chain polymerization, and the conversion of intracellular insoluble pectin to soluble pectin [27]. In tomato fruits, exogenous MT treatment decreases the protopectin while increasing the soluble pectin and promoting fruit softening [28]. In this study, exogenous MT treatment increased the cellulose content in plum fruits in the early fruit development and decreased the cellulose, ion-bound pectin, covalently-bound pectin, and hemicellulose contents in the later stage of fruit development. Exogenous MT also increased the water-soluble pectin content in plum fruits later in fruit development. These results imply that exogenous MT treatment promoted the conversion of insoluble pectin to soluble pectin, and the depolymerization of cellulose and hemicellulose accelerated the cell wall degradation and promoted the plum fruit ripening, consistent with previous findings [28].

The cell wall-modifying hydrolases play an important role in fruit texture changes. Notably, the synergy of different hydrolases catalyzes cell wall degradation, a complex physiological process [29]. For example, PME catalyzes pectin demethylation, promoting the conversion of pectin lipids to pectic acid, which provides substrates for PG to act on and promote fruit softening [10]. β-GAL acts on pectin and hemicellulose, reducing the galactose residues in fruits and causing cell wall structural instability [30]. In synergy with XTH, xyloglucan, the most abundant hemicellulose in the plant cell wall, catalyzes the hydrolysis and transfer of xyloglucan to achieve cell wall remodeling [31]. In this study, exogenous MT treatment increased the plum fruit and pulp hardness 21 days post-treatment. Exogenous MT treatment also decreased the PL, α-AF, and EG activities and increased the XTH activity in plum fruits in the early development stage. These results indicate that the cell wall hydrolysis-related enzyme activities were decreased after exogenous MT treatment at the early stages of fruit development.

On the other hand, XTH, a bifunctional enzyme, mainly played an integrative role in plum fruits, integrating newly secreted xyloglucan chains into the cell wall and participating in the formation of a complex xyloglucan–cellulose structure, ultimately increasing fruit hardness 21 days after exogenous MT treatment. Only 2 of the 10 *XTH* genes are associated with tomato fruit ripening in tomato fruits. Moreover, XET and XEH activities are higher during fruit development and may mainly play a role in maintaining the structural integrity of the cell wall. Thus, the down-regulation of *XTH* during fruit ripening may facilitate fruit softening [32]. In this study, exogenous MT treatment decreased the plum fruit hardness, pulp hardness, and fruit elasticity during maturity (28 days after treatment). At the same time, the PME, PG, PL, β-GAL, and α-AF activities in the exogenous MT-treated plum fruits were higher than the control and the XTH and EG activities were lower. These results revealed that the exogenous MT treatment increased the pectin hydrolysis-related enzyme activities, causing the plum fruit to soften during the fruit ripening. However, previous studies revealed that MT treatment inhibits the activities of PME, PG, β-GAL, and Cx in fruits at the post-harvest storage stage [33]. This may be due to the different mechanisms of MT action at different treatment periods; thus, further studies must be conducted.

In the present study, the DEGs in MT-treated plum fruits regulated the pentose and glucuronate interconversions, phenylpropanoid biosynthesis, cyanoamino acid metabolism, and galactose metabolism pathways. These four pathways are mainly regulated by genes related to cell wall metabolism. A previous study revealed that the cell wall metabolism-related genes in jujube fruits treated with MT are enriched in the interconversion of pentose and glucuronide and galactose metabolism pathways [34]. Transcriptome analysis has also revealed that the rapid decline in strawberry fruit hardness during fruit ripening is regulated by cell wall metabolism-related DEGs. Furthermore, the cell wall modifying enzyme-related DEGs are important in strawberry fruit development and ripening [35]. In pear fruits, the transcript abundance of cellulose synthase genes, *XTH* and *Exps,* significantly changes with decreasing pear fruit hardness after ripening. Moreover, the high expression levels of genes encoding pectin-degrading enzymes induce an increase in the pectic oligomers, which favors the increase in ethylene yield [36]. Exogenous MT treatment up-regulates the cell wall structure-related genes, including *PG*, *pectin methyl esterase 1* (*PE1*), and *β-galactosidase* (*TBG4*) in tomato fruits, which accelerate pectin hydrolysis and promote fruit ripening and softening [28]. Many proteins involved in fruit ripening and cell wall metabolism are also affected by exogenous MT treatment. Furthermore, exogenous MT treatment positively regulates fruit ripening while negatively regulating fruit senescence [16]. In this study, 11 and 16 genes related to cell wall metabolism following exogenous MT treatment were differentially expressed in plum fruits at 21 and 28 days, respectively. Notably, *PPE8B*, *XTH23*, *At4g24780*, and *GSVIVT0002692001* in exogenous MT-treated plum fruits were up-regulated at 21 days, while *RCA* and *PME* were down-regulated. At the same time, *PGIP*, *Xyl2*, *BGLU41*, and *BGLU11* were up-regulated at 28 days in the exogenous MT-treated plum fruits, while PME activity inhibitor, *PPE8B*, *RCA*, *BGAL8*, *XTH23*, and *PECS-1.1* were down-regulated. This implies that the exogenous MT treatment mediated the conversion of ion-bound and covalently-bound pectin to water-soluble pectin and the degradation of cellulose and hemicellulose through the pentose and glucuronate interconversions, phenylpropanoid biosynthesis, cyanoamino acid metabolism, and galactose metabolism pathways at the mature stage of plum fruits, thus promoting the fruit softening.

## 4. Materials and Methods

### 4.1. Materials

The Chinese plum variety ‘Qiangcuidali’ grafted on a wild peach rootstock was used in this study. The grafted plum seedlings were planted in a 2000 m^2^ orchard at Sichuan Huijia Seven Colorful Fields Agricultural Technology Co., Chengdu, China (30°38′ N, 104°13′ E) in February 2018. The seedling spacing was 4 m (plant space) × 3 m (row space). MT was obtained from Beijing Solarbio Science & Technology Co., Ltd. (Beijing, China).

### 4.2. Experimental Design

In May 2022, when the plum fruits were at the second rapid expansion period (90 days after flowering), 18 plum trees were selected for the present study. Among them, nine plum trees were treated with 100 μmol/L of MT [11,33], and the other nine were treated with tap water to serve as the control. Briefly, MT was dissolved in tap water and sprayed on the leaves until it started dripping (about 3 L per tree). The controls were sprayed with the same volume of tap water. Each treatment was replicated thrice with three trees per replicate. The MT and tap water treatments were applied again after 7 days. The fruits were sampled at 0, 7, 14, 21, and 28 (maturity) days after the first treatment. The fruit sampling at 0 and 7 days post-treatment was conducted before the spray application. Each time, 27 fruits were collected from the four directions per tree. Subsequently, the fruits were placed in an ice box and transported to the laboratory. In the laboratory, 12 fruits per tree were used for texture analysis, while the remaining 15 were immediately stored at −80 °C awaiting other index analyses.

### 4.3. Determination of Items

#### 4.3.1. Determination of the Fruit Texture

The plum fruit flesh was cut with a 10 mm diameter punch, and a 9 mm-thick section of pulp below the peel was retained. Next, the texture profile analysis (TPA) was performed using a TA/2R disc probe attached to a TA.XTC-18 texture analyzer (Shanghai Bosin Industrial Development Co., Ltd., Shanghai, China). The test type was downward pressure, the test target was displacement, the test depth was 6 mm, and the test speed was 0.8 mm/s to determine the elasticity of the fruits [37]. Subsequently, puncture analysis of the equatorial fruit parts was performed using a TA/2N needle-type probe to determine the fruit and pulp hardness. The test type was downward pressure, the test target was deformation, the deformation index was 20%, and the test speed was 2 mm/s.

#### 4.3.2. Determination of the Cell Wall Substance Content

The samples were dried and passed through a 0.425 mm sieve. Next, 1 g per sample was weighed in a centrifuge tube, and 30 mL of an 80% ethanol solution was added before boiling for 25 min. Subsequently, the precipitate was washed 1–2 times with 30 mL of 80% ethanol after cooling. The samples were then filtered under vacuum, and the residue was rinsed with 30 mL of the 80% ethanol solution. The obtained filter residue was soaked with 30 mL of 90% dimethyl sulfoxide overnight. Subsequently, the residue was soaked in acetone for 10–20 min to remove the dimethyl sulfoxide, vacuum filtered fully, rinsed in three changes of 30 mL of acetone each time, and dried to a constant weight. The final weight obtained was the cell wall substance [38].

The supernatant containing water-soluble pectin was obtained by weighing 0.05 g of the cell wall substance in a 10 mL capped centrifuge tube, adding 5 mL of deionized water, and shaking for 6 h. Next, 5 mL of sodium acetate buffer was added to the residue to obtain the supernatant containing ion-bound pectin after 6 h of shaking. Subsequently, 5 mL of a sodium carbonate solution was added to the residue to obtain the supernatant containing covalently-bound pectin after 6 h of shaking [39]. Finally, the carbazole ethanol colorimetric method described by Cao et al. (2007) [40] determined the different forms of pectin.

For hemicellulose quantification, 5 mL of a sodium hydroxide solution was added to the extracted pectin residue and shaken for 6 h to obtain the supernatant containing hemicellulose. Next, 2 mL of the hemicellulose supernatant was mixed with 3 mL of sulfuric acid and hydrolyzed in a boiling water bath at 100 °C for 2 h to obtain the hydrolyzed hemicellulose and the extracted hemicellulose residue (cellulose). Subsequently, 1.5 mL of sulfuric acid was added to the residue and left for 2 h. An appropriate amount of activated carbon and 3 mL of ultrapure water was then added to the residue and hydrolyzed in a boiling water bath at 100 °C for 2 h. The residue was then filtered to obtain the cellulose-containing solution. Finally, the cellulose and hemicellulose contents were determined by the anthrone and ethyl acetate colorimetric method [41].

The lignin content was determined by the acetylation method using the lignin content determination kit (Suzhou Grace Biotechnology Co., Ltd., Suzhou, China) following the manufacturer’s instructions.

#### 4.3.3. Determination of the MT Content

The MT content was determined by the enzyme-linked immunosorbent assay using the plant MT ELISA kit (Jiangsu Enzyme Immunoassay Industry Co., Ltd., Yancheng, China) following the manufacturer’s instructions.

#### 4.3.4. Determination of the Cell Wall Metabolism-Related Enzyme Activity

PME, pectin lyase (PL), PG, β-galactosidase (β-GAL), xyloglucan endo glycosyltransferase/hydrolase (XTH), α-L-arabinofuranosidase (α-AF), and endo-β-1,4-glucanase (EG) activities were determined using the PME, PL, PG, β-GAL, XTH, α-AF, and EGase kits (Suzhou Grace Biotechnology Co., Ltd., Suzhou, China), respectively, following the manufacturer’s instructions.

#### 4.3.5. RNA Library Preparation and Sequencing

RNA was extracted from the fruit pulp using the RNAprep Pure Plant Kit (Tiangen, Beijing, China) according to the instructions provided by the manufacturer. Next, the extracted RNA integrity was assessed using the RNA Nano 6000 Assay Kit on the Agilent Bioanalyzer 2100 system (Agilent Technologies, Santa Clara, CA, USA) according to the manufacturer’s protocol. RNA sequencing libraries were then generated using the Hieff NGS Ultima Dual-mode mRNA Library Prep Kit for Illumina (Yeasen Biotechnology (Shanghai) Co., Ltd., Shanghai, China) following the manufacturer’s instructions.

#### 4.3.6. Data Processing and Analysis of Differentially Expressed Genes (DEGs)

After sequencing, the obtained raw data (raw reads) were processed to obtain high-quality clean data. Next, sequence comparison to the reference genome of plum was performed using the Hisat2 tools. Subsequently, the differential expression analysis between treatments was performed using DESeq2. Genes with an adjusted *p* < 0.01 and fold change ≥ 2 were denoted as differentially expressed. Finally, the gene ontology (GO) and the Kyoto Encyclopedia of Genes and Genomes (KEGG) enrichment analyses of the DEGs were performed.

#### 4.3.7. Candidate DEGs Expression Analysis by Quantitative RT-PCR (qRT-PCR)

The first strand cDNA synthesis of the selected DEGs was performed using the M5 Super plus qPCR RT kit with gDNA remover from Mei5 Biotechnology Co., Ltd., Beijing, China, referring to the manufacturer’s instructions. The primers were designed using Primer Premier 5.0 software according to the reference gene sequences in plums (Table 4). RT-qPCR of the DEGs was performed on the CFX96TM Real-Time System platform using the 2X M5 HiPer SYBR Premix EsTaq (with Tli RNaseH) kit from Mei5 Biotechnology Co., Ltd., with CAC as the internal reference gene. The relative DEG expression levels were calculated using the 2^−ΔΔCT^ method [42].

### 4.4. Statistical Analysis

The data were analyzed in triplicate by Student’s *t*-test (0.01 ≤ *p* < 0.05 or *p* < 0.01) using the SPSS 20.0.0 software (IBM, Inc., Armonk, NY, USA).

## 5. Conclusions

Exogenous MT treatment increases the endogenous MT in plum fruits at the early stage of fruit development. At maturity, exogenous MT treatment accelerates the plum fruit softening by decreasing the fruit hardness, promoting the conversion of ion-bound and covalently-bound pectin to water-soluble pectin, and degrading the cellulose and hemicellulose. Furthermore, exogenous MT treatment increases the PME, PG, PL, β-GAL, and α-AF activities and decreases XTH and EG activities during plum fruit ripening. Exogenous MT treatment also mediates the pentose and glucuronate interconversions, phenylpropanoid biosynthesis, cyanoamino acid metabolism, and galactose metabolism pathways, regulating the expression of genes related to cell wall metabolism.

## Figures and Tables

**Figure 1 ijms-24-13495-f001:**
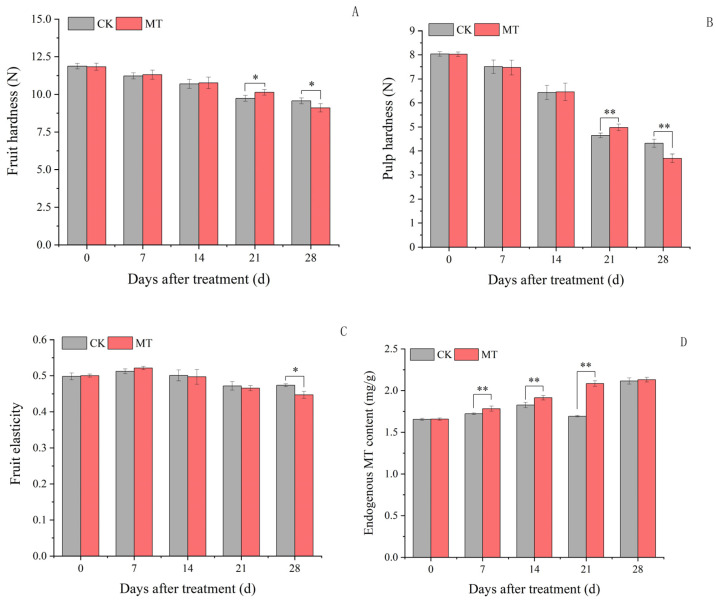
Texture, structure, and endogenous MT content of plum fruits. Values represent the mean ± SE (n = 3). Asterisks indicate significant differences between the treatments using the Student’s *t*-test (*: 0.01 ≤ *p* < 0.05; **: *p* < 0.01). (**A**): Fruit hardness; (**B**): pulp hardness; (**C**): fruit elasticity; (**D**): endogenous MT content.

**Figure 2 ijms-24-13495-f002:**
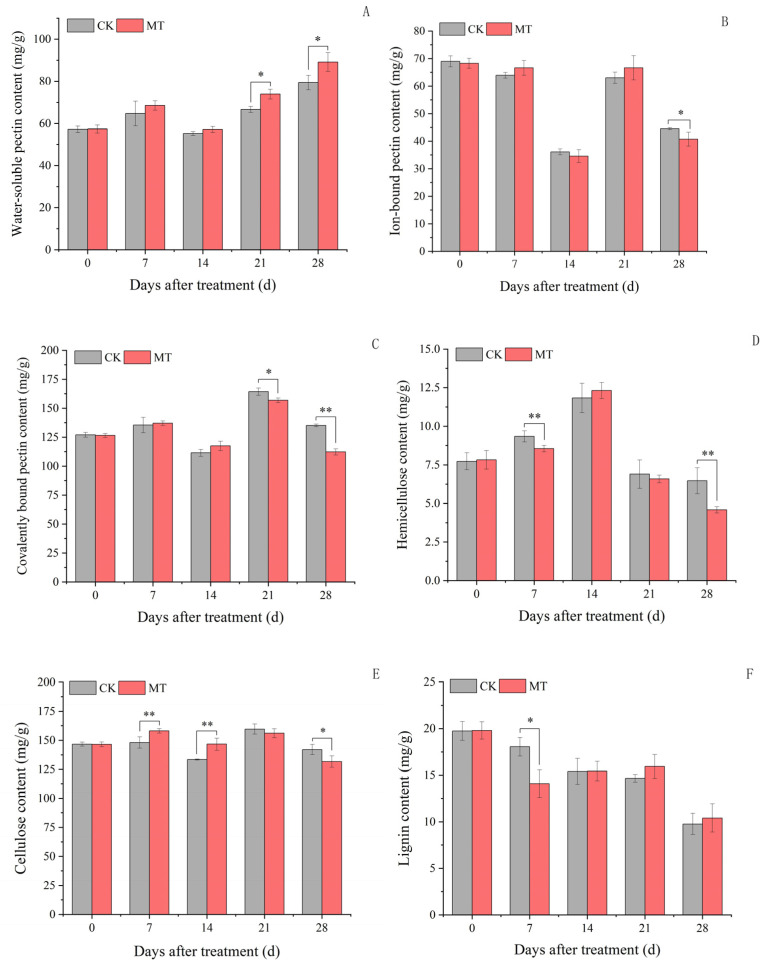
Cell wall substance content in plum fruits. Values represent the mean ± SE (n = 3). Asterisks indicate significant differences between the treatments using the Student’s *t*-test (*: 0.01 ≤ *p* < 0.05; **: *p* < 0.01). (**A**): Water-soluble pectin content; (**B**): ion-bound pectin content; (**C**): covalently-bound pectin content; (**D**): hemicellulose content; (**E**): cellulose content; (**F**): lignin content.

**Figure 3 ijms-24-13495-f003:**
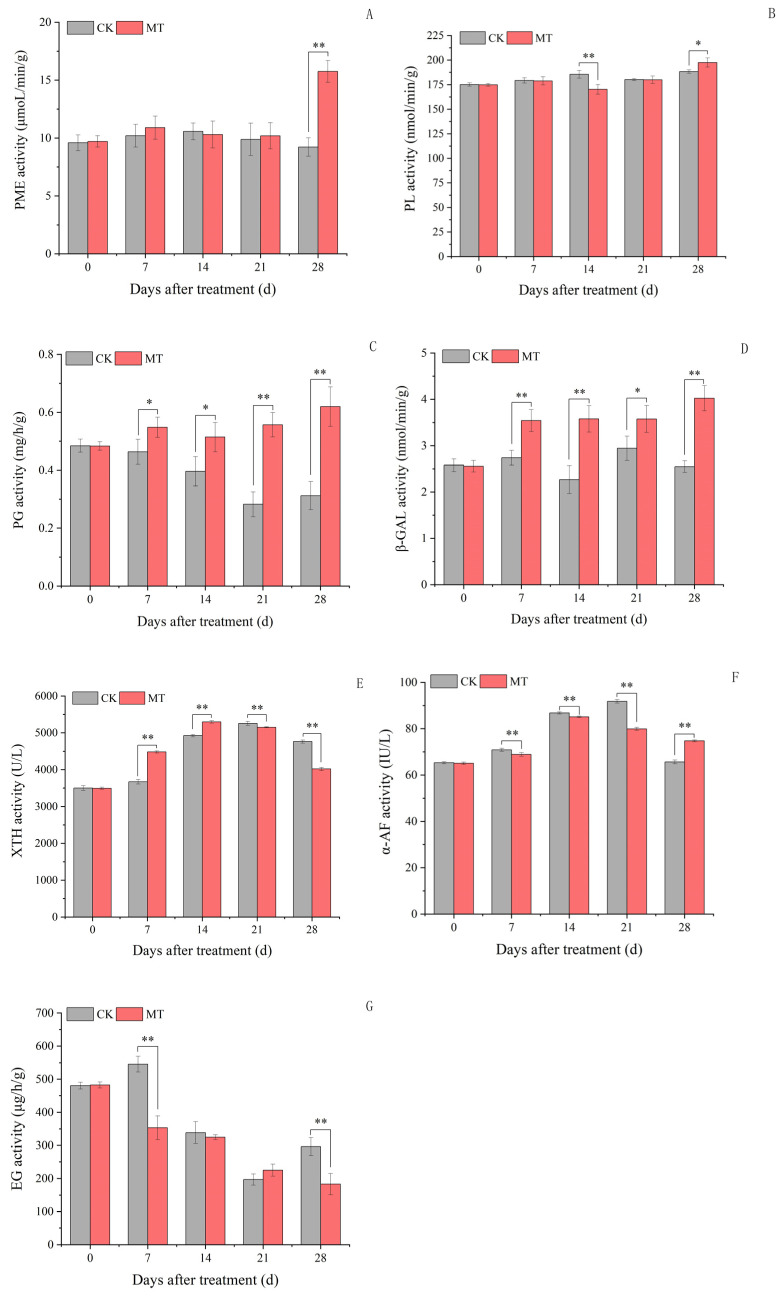
Cell wall metabolism-related enzyme activity of plum fruits. Values represent the mean ± SE (n = 3). Asterisks indicate significant differences between the treatments using the Student’s *t*-test (*: 0.01 ≤ *p* < 0.05; **: *p* < 0.01). (**A**): Pectin methyl esterase (PME) activity; (**B**): pectin lyase (PL) activity; (**C**): polygalacturonase (PG) activity; (**D**): β-galactosaminidase (β-GAL) activity; (**E**): xyloglucan endotransglycosylase/hydrolase (XTH) activity; (**F**): α-L-arabinofuranosidase (α-AF) activity; (**G**): endo-β-1,4-glucanase (EG) activity.

**Figure 4 ijms-24-13495-f004:**
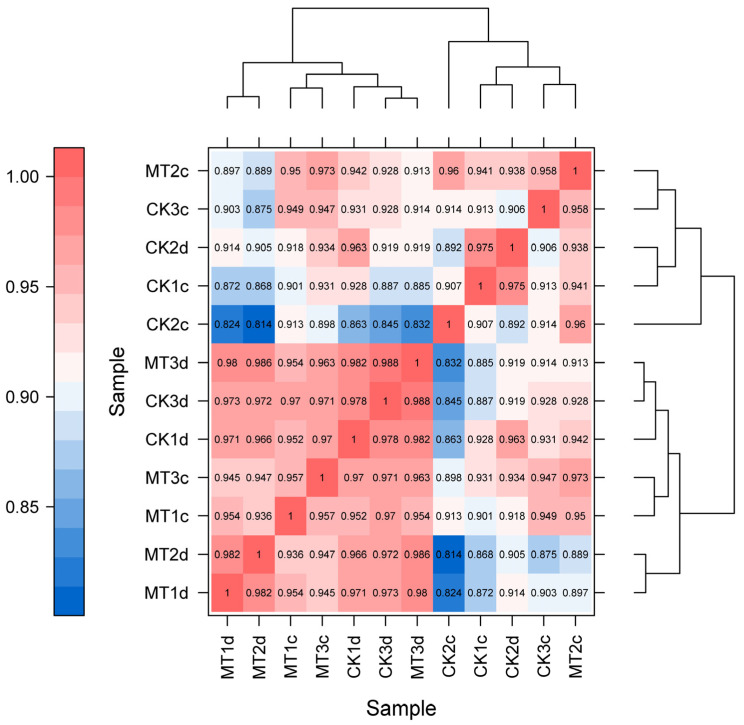
Heatmap of gene expression correlation of samples. CK1c, CK2c, and CK3c represent three replicates of CK at 21 days after treatment. CK1d, CK2d, and CK3d represent three replicates of CK at 28 days after treatment. MT1c, MT2c, and MT3c represent three replicates of MT at 21 days after treatment. MT1d, MT2d, and MT3d represent three replicates of MT at 28 days after treatment.

**Figure 5 ijms-24-13495-f005:**
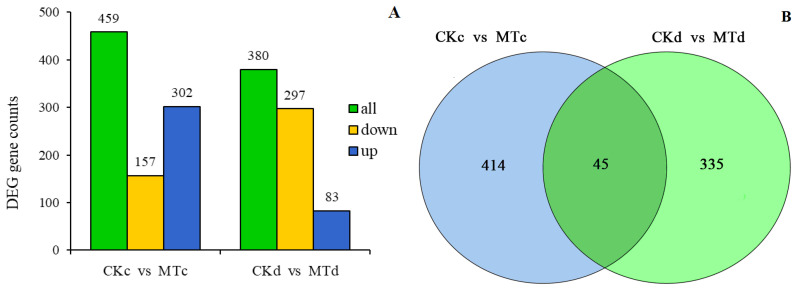
Statistical diagram of DEGs. (**A**): Number of DEGs; (**B**): Venn diagram of DEGs. CKc and MTc represent 21 days after treatment. CKd and MTd represent 28 days after treatment.

**Figure 6 ijms-24-13495-f006:**
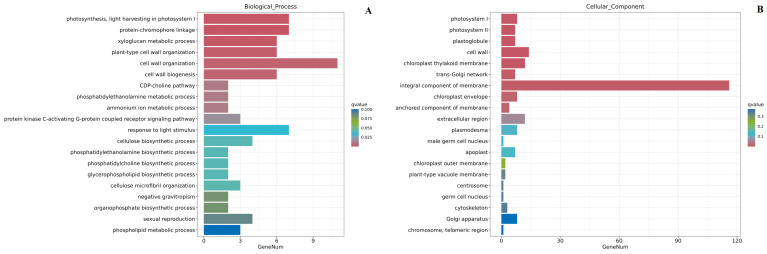
GO enrichment analysis of DEGs at 21 and 28 days after treatment. (**A**): Biological process in c period (21 days after treatment); (**B**): cellular component in c period (21 days after treatment); (**C**): molecular function in c period (21 days after treatment); (**D**): biological process in d period (28 days after treatment); (**E**): cellular component in d period (28 days after treatment); (**F**): molecular function in d period (28 days after treatment).

**Figure 7 ijms-24-13495-f007:**
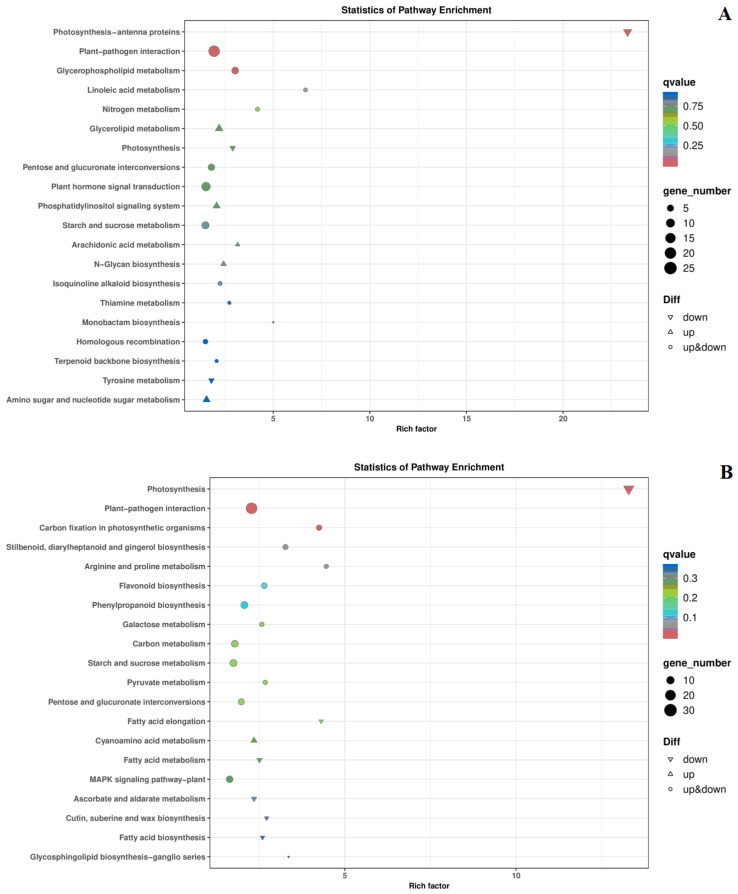
KEGG enrichment analysis of DEGs at 21 and 28 days after treatment. (**A**): in c period (21 days after treatment); (**B**): in d period (28 days after treatment).

**Figure 8 ijms-24-13495-f008:**
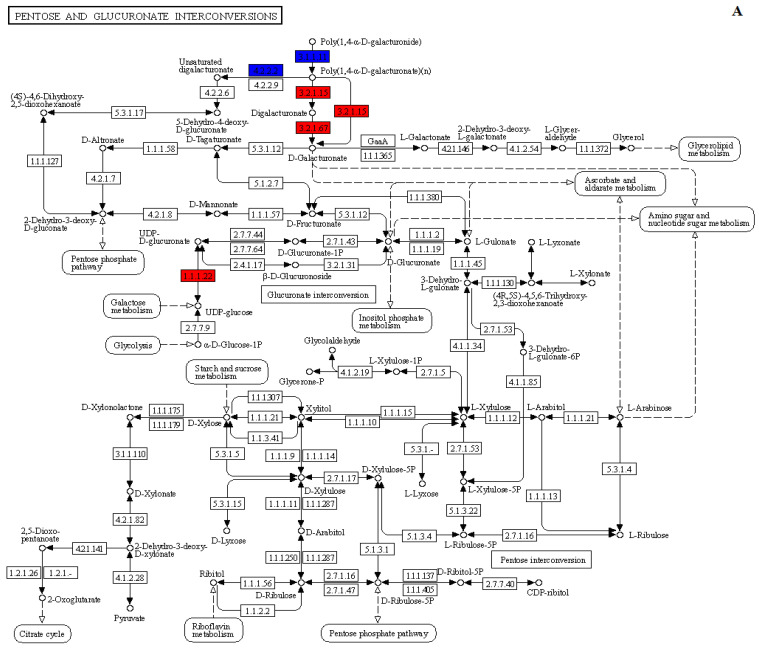
KEGG pathway annotation map of DEGs. (**A**): DEGs of the pentose and glucuronate interconversions pathway in period c (21 days after treatment); (**B**): DEGs of the phenylpropanoid biosynthesis pathway in period c (21 days after treatment); (**C**): DEGs of the pentose and glucuronate interconversions pathway in period d (28 days after treatment); (**D**): DEGs of the phenylpropanoid biosynthesis pathway in period d (28 days after treatment); (**E**): DEGs of the cyanoamino acid metabolism pathway in period d (28 days after treatment); (**F**): DEGs of the galactose metabolism pathway in period d (28 days after treatment). Red represents the up-regulation expressions of genes, green represents down-regulation expressions of genes, and blue represents both up-regulation and down-regulation expressions of genes.

**Figure 9 ijms-24-13495-f009:**
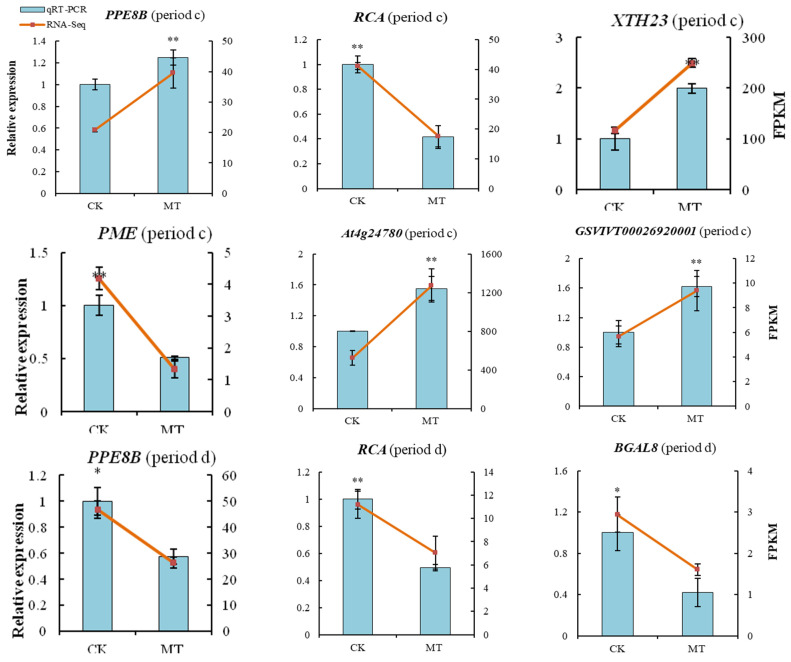
Relative expressions of screened DEGs. Values represent the mean ± SE (*n* = 3). Asterisks indicate significant differences between the treatments using the Student’s *t*-test (*: 0.01 ≤ *p* < 0.05; **: *p* < 0.01). Period c: 21 days after treatment; period d: 28 days after treatment.

**Table 1 ijms-24-13495-t001:** Statistical table of transcriptome sequencing data.

Samples	Clean Reads	Clean Bases	GC Content	Percentage ≥ Q30
CK1c	24,047,569	7,195,641,968	46.25%	94.31%
CK2c	22,524,544	6,738,533,176	45.96%	94.35%
CK3c	22,254,407	6,658,967,860	45.96%	94.39%
CK1d	22,766,320	6,811,703,926	46.03%	94.57%
CK2d	19,728,390	5,892,366,136	46.27%	94.89%
CK3d	22,134,960	6,623,316,176	46.08%	94.47%
MT1c	22,571,792	6,754,964,140	46.13%	94.27%
MT2c	21,401,071	6,404,898,452	45.95%	94.61%
MT3c	21,720,819	6,501,062,054	45.94%	94.62%
MT1d	24,216,209	7,245,015,474	46.14%	94.45%
MT2d	22,413,674	6,708,954,516	46.04%	94.20%
MT3d	21,038,544	6,297,492,174	45.95%	95.00%

K1c, CK2c, and CK3c represent three replicates of CK at 21 days after treatment. CK1d, CK2d, and CK3d represent three replicates of CK at 28 days after treatment. MT1c, MT2c, and MT3c represent three replicates of MT at 21 days after treatment. MT1d, MT2d, and MT3d represent three replicates of MT at 28 days after treatment.

**Table 2 ijms-24-13495-t002:** Statistics of sequence comparison of sample sequencing data with the selected reference genome.

Sample	Total Reads	Mapped Reads	Uniq Mapped Reads	Multiple Map Reads	Reads Map to ‘+’	Reads Map to ‘−’
CK1c	48,095,138	45,248,937 (94.08%)	40,854,635 (84.95%)	4,394,302 (9.14%)	26,672,296 (55.46%)	26,685,363 (55.48%)
CK2c	45,049,088	42,402,828 (94.13%)	39,140,794 (86.88%)	3,262,034 (7.24%)	23,701,056 (52.61%)	23,697,471 (52.60%)
CK3c	44,508,814	41,746,240 (93.79%)	38,631,674 (86.80%)	3,114,566 (7.00%)	23,243,929 (52.22%)	23,248,047 (52.23%)
CK1d	45,532,640	42,705,134 (93.79%)	39,323,894 (86.36%)	3,381,240 (7.43%)	24,101,914 (52.93%)	24,108,373 (52.95%)
CK2d	39,456,780	37,007,767 (93.79%)	33,478,908 (84.85%)	3,528,859 (8.94%)	21,746,347 (55.11%)	21,749,992 (55.12%)
CK3d	44,269,920	41,299,332 (93.29%)	38,207,075 (86.30%)	3,092,257 (6.99%)	23,034,074 (52.03%)	23,038,665 (52.04%)
MT1c	45,143,584	42,433,505 (94.00%)	39,213,367 (86.86%)	3,220,138 (7.13%)	23,684,264 (52.46%)	23,695,294 (52.49%)
MT2c	42,802,142	40,335,705 (94.24%)	37,342,807 (87.25%)	2,992,898 (6.99%)	22,441,527 (52.43%)	22,441,944 (52.43%)
MT3c	43,441,638	40,880,319 (94.10%)	37,895,016 (87.23%)	2,985,303 (6.87%)	22,695,312 (52.24%)	22,700,036 (52.25%)
MT1d	48,432,418	45,536,663 (94.02%)	42,110,064 (86.95%)	3,426,599 (7.08%)	25,405,517 (52.46%)	25,417,786 (52.48%)
MT2d	44,827,348	42,115,270 (93.95%)	38,933,242 (86.85%)	3,182,028 (7.10%)	23,467,777 (52.35%)	23,486,024 (52.39%)
MT3d	42,077,088	39,808,166 (94.61%)	36,812,988 (87.49%)	2,995,178 (7.12%)	22,225,118 (52.82%)	22,234,922 (52.84%)

K1c, CK2c, and CK3c represent three replicates of CK at 21 days after treatment. CK1d, CK2d, and CK3d represent three replicates of CK at 28 days after treatment. MT1c, MT2c, and MT3c represent three replicates of MT at 21 days after treatment. MT1d, MT2d, and MT3d represent three replicates of MT at 28 days after treatment.

**Table 3 ijms-24-13495-t003:** Statistics on the number of annotated DEGs.

DEG Set	Total	COG	GO	KEGG	KOG	NR	Pfam	Swiss-Prot	eggNOG
CKc vs. MTc	436	123	350	302	174	436	353	329	396
CKd vs. MTd	369	128	306	260	154	367	297	283	349

CKc and MTc represent 21 days after treatment. CKd and MTd represent 28 days after treatment.

**Table 4 ijms-24-13495-t004:** Primer information of qRT-PCR.

Gene Name	Gene ID in NCBI (NR_Symbol)	Description	Forward Primer Sequence (5′-3′)	Reverse Primer Sequence (5′-3′)	Tm (°C)
*PPE8B*	LOC103340502	Pectinesterase/pectinesterase inhibitor PPE8B [*Prunus mume*]	TTTCCGACTGCCTTGATT	TTGCCCTTCTGATTCTGG	52.30
*RCA*	LOC117618678	Ribulose bisphosphate carboxylase/oxygenase activase, chloroplastic isoform X1 [*Prunus dulcis*]	GCACCGCTGAGCCTAAAT	TTCCACCTCTGCTACAATCCTG	57.17
*XTH23* (period c)	LOC18792321	Probable xyloglucan endotransglucosylase/hydrolase protein 23 [*Prunus persica*]	GCTTCTTACGCTGTCCCT	GCTCCAATCTGCCTTCAC	54.36
*PME*	LOC18789392	21 kDa protein [*Prunus persica*]	GCCGCTCTTCACGACTGCT	CATTCTCCGCTCGCTTGGT	60.43
*At4g24780*	-	Pectase lyase [*Prunus salicina*]	GCAGAGGCTGGCAGATTG	CGACCGTCGATGGTCTTG	56.85
*GSVIVT00026920001*	LOC103323077	Probable polygalacturonase [*Prunus mume*]	TGGTGGGATTGGTTTAGC	ATAAGGCGACTCTGGAGG	52.59
*BGAL8*	LOC110771136	Beta-galactosidase 8-like isoform X1 [*Prunus avium*]	TGGAACAGGAAACGGTAA	CTGAAGCCCAACAGTCAA	51.31
*PGIP*	-	Polygalacturonase inhibiting protein [*Prunus salicina*]	CCTCCTCTGCTTGACCCT	GGAGTTGATGCGGTTTGT	55.86
*XTH23* (period d)	-	Unnamed protein product [*Prunus armeniaca*]	CAAAGAGCAGCAGTTCTACCT	GCCCAGTCATCAGCGTTC	56.76
*PECS-1.1*	LOC103327551	Pectinesterase [*Prunus mume*]	GACTTGCCTTGATGGGTTCT	GCATTGCTGCATAGTTGTTCT	56.05
*Xyl2*	LOC103331025	Beta-glucosidase BoGH3B isoform X1 [*Prunus mume*]	TTTGAGAACCCTTTGGCTGAT	TTGGGAAGAGGTATGACTGGAT	55.56
*BGLU41*	-	Unnamed protein product [*Prunus armeniaca*]	AAGTACCAGAACCCTCCG	AAACCGAACAGTGTAGCC	52.26
*BGLU11*	LOC103330636	Beta-glucosidase 11-like [*Prunus mume*]	GGACCTGTCAACCCGAAGG	CGTTGTGGCGGAAGAAAT	54.87
*CAC*	At1g60780	Clathrin adaptor complexes medium subunit	GGGATACGCTACAAGAAGAATGAG	CTTACACTCTGGCATACCACTCAA	58.65

Period c: 21 days after treatment; period d: 28 days after treatment.

## Data Availability

The data presented in this study are available on request from the corresponding author.

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
