# Peer review of "Transcriptome Profiles Reveal the Promoting Effects of Exogenous Melatonin on Fruit Softening of Chinese Plum"

_ijms, 2023, doi:10.3390/ijms241713495_

Round 1
Reviewer 1 Report
The manuscript by Li et al. describes the effect of exogenous melatonin on processes in Chinese plum fruits during ripening. The paper could be published after revision.
L. 61. “Plum (Prunus salicina Lindl.) is a common temperate fruit…”; L. 74. “The plum variety ‘Qiangcuidali’…”. Plum is the common name for Prunus domestica (European plum), and the common name for Prunus salicina is Chinese plum or Japanese plum. The common name should be changed.
L. 61-63. “Plum (Prunus salicina Lindl.) is a common temperate fruit … but the content of lignin in fruits affects its commercial value (Radović et al., 2020).” This reference should be replaced because Radović et al. studied European plum cultivars and did not analyze the content of lignin in fruits.
L. 164, 255, 257, etc. Suppl. Tables and Figures are absent.
The quality of Figure 5 should be improved.
Fig. 5, 6. What does “period c” and “period d” mean?
There is no information about Author Contribultion, Funding, Suppl. Materials.
Author Response
The manuscript by Li et al. describes the effect of exogenous melatonin on processes in Chinese plum fruits during ripening. The paper could be published after revision.
RESPONSE: Thank you for your reviewing.
- 61. “Plum (Prunus salicina Lindl.) is a common temperate fruit…”; L. 74. “The plum variety ‘Qiangcuidali’…”. Plum is the common name for Prunus domestica (European plum), and the common name for Prunus salicina is Chinese plum or Japanese plum. The common name should be changed.
RESPONSE: We have revised as “Chinese plum” in title, abstract, L. 61, L. 70, and L. 74.
- 61-63. “Plum (Prunus salicina Lindl.) is a common temperate fruit … but the content of lignin in fruits affects its commercial value (Radović et al., 2020).” This reference should be replaced because Radović et al. studied European plum cultivars and did not analyze the content of lignin in fruits.
RESPONSE: We have replaced it as “Qiu, X.; Zhang, H.; Zhang, H.; Duan, C.; Xiong, B.; Wang, Z. Fruit textural characteristics of 23 plum (Prunus salicina Lindl) cultivars: Evaluation and cluster analysis. Hortscience 2021, 56, 816-823.” And “He, M.; Wu, Y.; Wang, Y.; Hong, M.; Li, T.; Deng, T.; Jiang, Y. Valeric acid suppresses cell wall polysaccharides disassembly to maintain fruit firmness of harvested ‘Waizuili’ plum (Prunus salicina Lindl). Sci. Hortic. 2022, 291, 110608.”
- 164, 255, 257, etc. Suppl. Tables and Figures are absent.
RESPONSE: We have added the Suppl. Tables and Figures in the text, and renumbered the figures and tables.
The quality of Figure 5 should be improved.
RESPONSE: We have revised.
Fig. 5, 6. What does “period c” and “period d” mean?
RESPONSE: Period c: 21 days after treatment; period d: 28 days after treatment. We have added in all responding figures and tables.
There is no information about Author Contribultion, Funding, Suppl. Materials.
RESPONSE: We have added Author Contribultion and Funding, and there is no Suppl. Materials.
Reviewer 2 Report
The authors have provided a very insightful and in-depth evaluation of how the cell wall composition can affect fruit ripening, with specific focus on melatonin. The authors have also done a good job in demonstrating the molecular mechanisms of this effect, using comprehensive analysis tools (ELISA, enzyme activity and transcriptomics.)
The authors could provide some more explanation of the impacts of their findings, to cater to a broader audience. Are there melatonin (or other exogenous) treatment modalities currently used that could harness this effect in plum fruit growth?
Is this translatable to metabolic effects seen in human cells/ or human patients? Out of curiosity, what are the parallels?
Author Response
The authors have provided a very insightful and in-depth evaluation of how the cell wall composition can affect fruit ripening, with specific focus on melatonin. The authors have also done a good job in demonstrating the molecular mechanisms of this effect, using comprehensive analysis tools (ELISA, enzyme activity and transcriptomics.)
RESPONSE: Thank you for your reviewing.
The authors could provide some more explanation of the impacts of their findings, to cater to a broader audience. Are there melatonin (or other exogenous) treatment modalities currently used that could harness this effect in plum fruit growth?
RESPONSE: There are no reports of melatonin (or other exogenous) treatment applying during the plum fruit growth to promote plum fruit softening or ripening. The effects of MT on the ripening of horticultural crops such as tomatoes, strawberries, and grapes have been reported. We have added more information to explanation of the impacts of their findings in discussion section.
Is this translatable to metabolic effects seen in human cells/ or human patients? Out of curiosity, what are the parallels?
RESPONSE: In vertebrates, MT is synthesized in the pineal gland in four consecutive enzymatic reactions with tryptophan as the precursor. In human body, MT plays a key role in regulating the biological clock and the sleep-wake cycle, and also has the anti-inflammatory, antioxidant, immunomodulatory, analgesic, and anxiety-reducing properties. In addition, MT is an antioxidant that neutralizes free radicals and reduces oxidative stress. This helps protect cells from oxidative damage and fights the development of chronic diseases. Some research suggests that MT may have an effect on the immune system, modulating immune cell activity and immune response. So, the effects of MT on human cells/ or human patients are completely different from plants. We have more information about the effects of MT on human body in introduction section.